# Leaf Traits of Trees in Tropical Dry Evergreen Forests of Peninsular India

**Muthulingam Udayakumar** [1],* and **Thangavel Sekar** [2]

1   Department of Plant Science, Manonmaniam Sundaranar University, Abishekapatti, Tirunelveli 627012, Tamil Nadu, India
2   PG and Research Department of Botany, Pachaiyappa's College, Chennai 600030, Tamil Nadu, India; tsekar_bot@yahoo.com
*   Correspondence: udayakumar@msuniv.ac.in

**Abstract:** A plant functional trait study was conducted to know the existing relationship between important leaf traits namely, specific leaf area (SLA), leaf dry matter content (LDMC), and leaf life span (LL) in tropical dry evergreen forest (TDEFs) of Peninsular India. Widely accepted methodologies were employed to record functional traits. The relationships between SLA and LDMC, LDMC and LL, and SLA and LL were measured. Pearson's coefficient of correlation showed a significant negative relationship between SLA and LDMC, and SLA and LL, whereas a significant positive relationship was prevailed between LDMC and LL. The mean trait values (SLA, LDMC, and LL) of evergreens varied significantly from deciduous species. SLA had a closer relationship with LDMC than LL. Similarly, LL had a closer relationship with SLA than LDMC. Species with evergreen leaf habits dominated forest sites under study. Evergreen species dominate the study area with a high evergreen-deciduous ratio of 5.34:1. The S strategy score of trees indicated a relatively higher biomass allocation to persistent tissues. TDEFs occur in low elevation, semiarid environment, but with the combination of oligotrophic habitat, high temperature and longer dry season these forests were flourishing as a unique evergreen ecosystem in the drier environment. The relationships found between leaf traits were in concurrence with earlier findings. Trees of TDEFs survive on the poor-nutrient habitat with a low SLA, high LDMC, and LL. This study adds baseline data on key leaf traits to plant functional trait database of India.

**Keywords:** functional traits; leaf dry matter content; leaf lifespan; specific leaf area; Indian forest

## 1. Introduction

One of the recent trends in plant ecological studies are to find plant traits capable of expressing differences in ecological behavior among species [1]. Classifying species into clusters based on their functional characteristics rather than on conventional methods promoted intensive research to find out various plant functional traits [2]. Plant traits represent specific functional adaptations to diverse biotic and abiotic factors, and they may act as valuable predictors of the response of species to various environmental circumstances [3]. Currently, creation of a large database for plant functional traits (PFTs) gets high priority in the research agenda of plant ecology since it helps in the understanding and prediction of the distribution of species in present and future environments [2,4].

Leaves are organs of great taxonomical and ecological importance [5]. Leaf habits were recognized as remarkable tools in ecological studies [6]; closely connected with growth and survival [1,2], they are widely considered as reliable predictors of plant performance and act as efficient indicators of resource-use strategies [7]. Specific leaf area (SLA) is regarded as a crucial factor in studies of plant productivity in the earlier twentieth century onwards [8,9]. Among many leaf traits, specific leaf area (SLA) and leaf dry matter content (LDMC) are reported to be better predictors of resource-use strategies of plants (plant-soil-climate interaction; [7,10]. LDMC was recognized as a cardinal trait along with SLA

and leaf lifespan (LL) in a global plant functional traits compilation study [11]. Further, assessing relationships among leaf traits and how traits differ between or among growth forms (herb, shrub, tree, and liana) and plant functional types (deciduous, evergreen) can give important insights into the selective pressures that shaped the evolution of the world's vegetation [12,13]. For example, in an infertile habitat, natural selection generally favors species with longer LL (higher nutrient residence time), which plays a positive role in nutrient conservation [14].

Leaf lifespan (LL) plays a fundamental role in the important tradeoff between plant growth rate, protection, and nutrient conservation [15]. Besides, leaf phenology and longevity possess the ability to reveal the relationship between resource availability and leaf habit [16]. According to Reich et al. [17], Williams–Linera [18] and Villar and Merino [19], information on mean leaf lifespan is a fundamental element in global analyses of leaf functional characters. Information on plant functional traits of tropical dry evergreen forest (TDEF) plants were limited. Recently, we found a significant negative relationship between leaf area and wood density with 56 TDEF tree species [20]. Information on leaf traits can facilitate for a better understanding of ecophysiology of TDEFs, hence, this study recorded data on important leaf traits viz. SLA, LDMC and LL from 44 tree species in ten TDEF sites. The study was conducted to estimate the degree of existing relationship between important leaf traits, namely, SLA-LDMC, SLA-LL, and LDMC-LL in selected forest sites which harbors core tree species of TDEF.

## 2. Materials and Methods

### 2.1. Study Area

This study was conducted in selected TDEFs located in Nagapattinam (NP), (10°10′ & 11°20′ N; 79°15′ & 79°50′ E) and Thiruvarur (TV), (10°20′ & 11°07′ N; 79°15′ & 79°45′ E) districts. These two districts situated in Tamil Nadu state are part of the Coromandel Coast. In Tamil Nadu, the Coromandel Coast is extended from Thiruvallur district in the north to Ramanathapuram district in the south. According to 2011 census the human population was 1.44 million and 1.16 million in Nagapattinam and Thiruvarur districts, respectively (https://www.tngov.in; accessed on 13 December 2011). The mean maximum and minimum annual temperature and rainfall is 32 °C, 24.60 °C, and 1174 mm, respectively; 36.9 °C, 29.8° C, and 1091 mm in Nagapattinam and Thiruvarur districts, respectively. The natural forest coverage of the Nagapattinam is 3557 ha and Thiruvarur is 2542 ha. Information about the occurrence of natural forest patches was obtained by frequent visits to villages and personal interviews with local people. Out of 24 forest sites (invariably all are sacred groves) identified, 10 sites (8 belonged to Nagapattinam district and 2 belonged to Thiruvarur district) where vegetation coverage exceeding one ha as in Ayakkaranpulam (AM), Andurkadu (AU), Jambavanodai (JI), Panndraththankadu (PK), Periyakuththagai (PI), Pushpavanam (PM), Thennampulam (TM), Theththakudi north (TN), Theththakudi south (TS), and Thillaivilagam (TV) were selected for the present ecological studies (as illustrated in Figure 1). Selected forest sites are typical representatives of TDEF [21], thus, they were chosen for plant functional trait studies.

The importance of tropical dry evergreen forest (TDEF) was initially recognized by Wimbush, who served as the chief conservator of forests of Madras Presidency. He writes in his classical paper on the forests of southern India as "On approaching the East Coast for instance in Chengleput district (Chengalpattu in Tamil), a type of forest is met which is semievergreen in character. Many of these species are quite different from there to the West, and although many individual trees are shrubs, they retain their leaves with the result that we never get completely leafless appearance so characteristic of true deciduous forests in hot weather" (as in [22]). In 1936, Champion applied the term TDEF in his classical and most important book titled 'Forest types of India'. The book was revised by Champion and Seth [23]. TDEFs occur as small patches of forest (0.5–10 ha) along the east coast of India from Visakapatnam (Andhra Pradesh) in the north to Ramanathapuram (Tamil Nadu) in the south. The total geographical cover of TDEF is 2072 km$^2$. The forest was classified as

an endangered forest type in India [21,24]. Sandy coast, interior coastal plains, and isolated hillocks of east coast region are the habitats of TDEF.

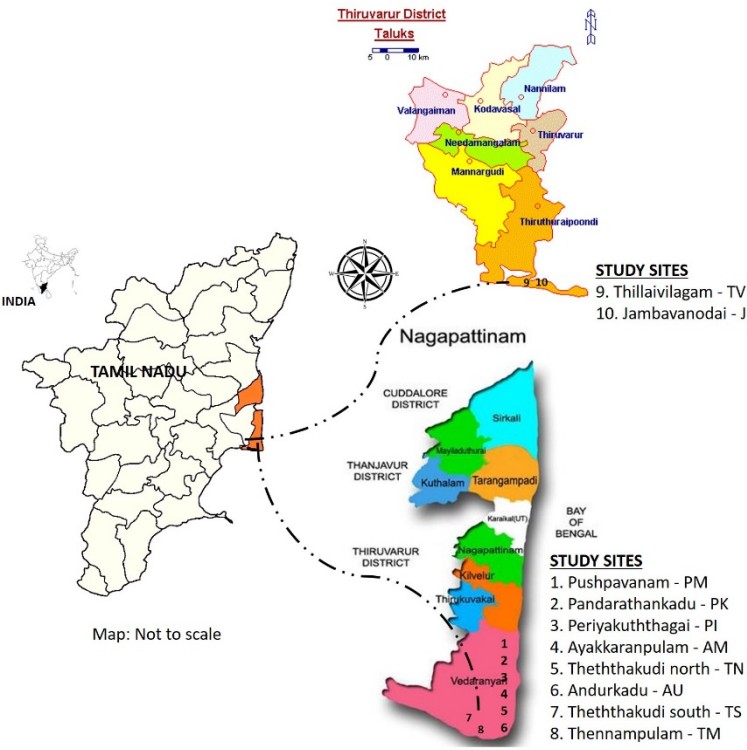

**Figure 1.** Map of study area showing locations of forest sites wherein leaf trait study was conducted.

### 2.2. Study Species

Tree species represented by ≥10 individuals in the study area were considered for leaf trait estimations. Leaf traits of 44 tree species, spread in 40 genera and 22 families, were measured. Most speciose families of study area are Rubiaceae (seven species) followed by Euphorbiaceae and Fabaceae (four each), while 11 families had single species each in the study area. A sum of 7816 trees ≥10 cm girth at breast height were measured in which 6583 are evergreens and 1233 are deciduous. Among 44 species, *Memecylon umbellatum* was represented by a large number of individuals (2336 trees), followed by *Garcinia spicata* (645), *Glycosmis mauritiana* (547), and *Canthium dicoccum* (393), whereas *Albizia lebbeck* had just 16 individuals in the study area (Table 1).

**Table 1.** Botanical name, family, and number of trees recorded in study area (exotic species appear in bold face).

| Botanical Name | Family | AM | AU | JI | PK | PI | PM | TM | TN | TS | TV | Study Area |
|---|---|---|---|---|---|---|---|---|---|---|---|---|
| *Aglaia elaeagnoidea* (A. Juss.) Benth. | Meliaceae | 2 | 2 | 1 | 2 | 28 | 3 | 2 | 4 | 2 | 1 | 47 |
| *Albizia lebbeck* (L.) Benth. | Fabaceae | 1 | 2 | 1 | 1 | 2 | 1 | 1 | 2 | 2 | 3 | 16 |
| *Allophylus serratus* (Hiern) Kurz | Sapindaceae | 2 | 2 | 1 | 1 | 4 | 3 | 12 | 2 | 17 | 5 | 49 |
| ***Anacardium occidentale* L.** | Anacardiaceae | 1 | 3 | 1 | 1 | 1 | 2 | 9 | 3 | 2 | 2 | 25 |
| *Atalantia monophylla* DC. | Rutaceae | 24 | 5 | 62 | 22 | 14 | 23 | 89 | 3 | 56 | 7 | 305 |
| *Azadirachta indica* A. Juss. | Meliaceae | 13 | 3 | 2 | 1 | 2 | 1 | 17 | 1 | 2 | 1 | 43 |
| *Benkara malabarica* (Lam.) Tirveng. | Rubiaceae | 13 | 16 | 2 | 20 | 26 | 11 | 43 | 69 | 46 | 10 | 256 |
| *Breynia vitis-idaea* (Burm. f.) C.E.C. Fisch. | Euphorbiaceae | 6 | 4 | 2 | 1 | 7 | 1 | 6 | 4 | 16 | 6 | 53 |
| *Canthium coromandelicum* Alston | Rubiaceae | 6 | 9 | 1 | 1 | 11 | 1 | 16 | 5 | 4 | 6 | 60 |
| *Canthium dicoccum* (Gaertn.) Merr. | Rubiaceae | 4 | 165 | 26 | 58 | 33 | 62 | 5 | 32 | 1 | 7 | 393 |
| *Cassia fistula* L. | Fabaceae | 9 | 8 | 2 | 51 | 30 | 15 | 3 | 5 | 9 | 8 | 140 |
| *Catunaregam spinosa* (Thunb.) Tirveng. | Rubiaceae | 13 | 2 | 3 | 1 | 1 | 2 | 2 | 4 | 11 | 2 | 41 |
| *Chionanthus zeylanicus* L. | Oleaceae | 3 | 10 | 5 | 4 | 2 | 5 | 3 | 4 | 3 | 2 | 41 |
| *Crateva magna* (Lour.) DC. | Capparaceae | 11 | 1 | 4 | 8 | 2 | 1 | 2 | 3 | 9 | 2 | 43 |
| *Diospyros ebenum* J. König | Ebenaceae | 4 | 14 | 3 | 4 | 99 | 2 | 1 | 34 | 5 | 1 | 167 |
| *Diospyros ferrea* (Willd.) Bakh. | Ebenaceae | 4 | 30 | 3 | 63 | 90 | 31 | 50 | 18 | 7 | 14 | 310 |
| *Diospyros montana* Roxb. | Ebenaceae | 5 | 38 | 4 | 5 | 3 | 2 | 1 | 4 | 2 | 4 | 68 |
| *Drypetes sepiaria* (Wight & Arn.) Pax & K. Hoffm. | Euphorbiaceae | 6 | 9 | 1 | 93 | 60 | 51 | 12 | 4 | 6 | 17 | 259 |

**Table 1.** *Cont.*

| Botanical Name | Family | Study Sites | | | | | | | | | | Study Area |
| | | AM | AU | JI | PK | PI | PM | TM | TN | TS | TV | |
|---|---|---|---|---|---|---|---|---|---|---|---|---|
| *Ehretia pubescens* Benth. | Boraginaceae | 18 | 3 | 2 | 3 | 1 | 9 | 5 | 3 | 13 | 4 | 61 |
| *Ficus benghalensis* L. | Moraceae | 2 | 2 | 8 | 3 | 1 | 1 | 1 | 2 | 3 | 3 | 26 |
| *Ficus hispida* L. f. | Moraceae | 7 | 2 | 12 | 3 | 4 | 2 | 1 | 3 | 2 | 7 | 43 |
| *Flacourtia indica* (Burm. f.) Merr. | Salicaceae | 1 | 5 | 2 | 2 | 11 | 2 | 4 | 3 | 9 | 5 | 44 |
| *Garcinia spicata* Hook. | Clusiaceae | 26 | 11 | 14 | 7 | 10 | 59 | 87 | 11 | 70 | 350 | 645 |
| *Glycosmis mauritiana* Tanaka | Rutaceae | 54 | 39 | 66 | 43 | 29 | 23 | 20 | 116 | 42 | 115 | 547 |
| *Gmelina asiatica* L. | Verbenaceae | 5 | 3 | 3 | 1 | 41 | 5 | 3 | 37 | 21 | 3 | 122 |
| *Ixora pavetta* Andrews | Rubiaceae | 10 | 4 | 3 | 52 | 5 | 24 | 3 | 2 | 8 | 8 | 119 |
| *Lannea coromandelica* (Houtt.) Merr. | Anacardiaceae | 10 | 7 | 2 | 4 | 27 | 9 | 4 | 5 | 6 | 8 | 82 |
| *Lepisanthes tetraphylla* (Vahl.) Radlk. | Sapindaceae | 22 | 3 | 24 | 8 | 14 | 12 | 25 | 3 | 10 | 10 | 131 |
| *Madhuca longifolia* (J. König ex L.) J.F. Macbr. | Sapotaceae | 9 | 1 | 2 | 2 | 1 | 2 | 3 | 10 | 5 | 3 | 38 |
| *Manilkara hexandra* (Roxb.) Dubard | Sapotaceae | 3 | 76 | 1 | 23 | 5 | 28 | 2 | 20 | 4 | 7 | 169 |
| *Maytenus emarginata* (Willd.) Ding Hou | Celastraceae | 6 | 17 | 4 | 40 | 10 | 7 | 3 | 6 | 9 | 11 | 113 |
| *Memecylon umbellatum* Burm. f. | Melastomataceae | 5 | 224 | 9 | 454 | 309 | 234 | 2 | 483 | 55 | 561 | 2336 |
| *Ochna serrata* L. | Ochnaceae | 1 | 14 | 7 | 2 | 36 | 1 | 1 | 2 | 2 | 1 | 67 |
| *Pamburus missionis* (Wight) Swingle | Rutaceae | 11 | 4 | 6 | 5 | 7 | 4 | 5 | 12 | 9 | 3 | 66 |
| *Pavetta indica* L. | Rubiaceae | 6 | 8 | 1 | 2 | 5 | 2 | 11 | 1 | 1 | 6 | 43 |
| *Phyllanthus reticulatus* Poir. | Euphorbiaceae | 5 | 1 | 3 | 2 | 1 | 2 | 1 | 1 | 1 | 5 | 22 |
| *Pongamia pinnata* (L.) Merr. | Fabaceae | 28 | 1 | 7 | 2 | 1 | 1 | 11 | 1 | 2 | 10 | 64 |
| *Pterospermum canescens* Roxb. | Malvaceae | 53 | 10 | 5 | 3 | 2 | 10 | 1 | 4 | 60 | 3 | 151 |
| *Sapindus emarginatus* Vahl | Sapindaceae | 1 | 2 | 2 | 11 | 1 | 2 | 2 | 5 | 2 | 2 | 30 |
| *Securenega leucopyrus* (Willd.) Muell.-Arg. | Euphorbiaceae | 2 | 3 | 1 | 4 | 2 | 2 | 1 | 1 | 11 | 5 | 32 |
| *Streblus asper* Lour. | Moraceae | 7 | 6 | 31 | 4 | 11 | 3 | 4 | 5 | 5 | 2 | 78 |
| *Syzygium cumini* (L.) Skeels | Myrtaceae | 6 | 3 | 32 | 1 | 1 | 1 | 45 | 44 | 2 | 28 | 163 |
| *Tamarindus indica* L. | Fabaceae | 2 | 3 | 1 | 2 | 4 | 1 | 14 | 2 | 1 | 2 | 32 |
| *Tarenna asiatica* Kuntze ex K. Schum. | Rubiaceae | 5 | 47 | 2 | 85 | 29 | 13 | 3 | 1 | 10 | 81 | 276 |
| Total | - | 432 | 822 | 374 | 1105 | 983 | 676 | 536 | 984 | 563 | 1341 | 7816 |

### 2.3. Specific Leaf Area and Leaf Dry Matter Content

Specific leaf area is the one-sided area of a fresh leaf divided by its oven-dry mass. Whole twig sections that had fully expanded and hardened leaves were collected from study area after 2 h of sunri25se. Samples were wrapped with moist paper, kept in plastic bags, and sealed to maintain them in water-saturated condition. Leaves of deciduous species were rehydrated in the dark by placing the cut end of the stem in deionized water for 6 h. Leaves (leaflets in the case of a compound leaf) were detached from the whole twig section just prior to leaf area measurement. Such removed leaf or leaflet was gently dried by using blotting paper before measurement. A total of 100 leaves (10 leaves each from ten individuals) along with petiole were measured to get leaf area with the help of Adobe Photoshop CS4 and HP Scanjet 2400 flat-bed scanner, Taikoo Shing, Hong Kong. Leaf samples collected from one individual each from all study sites. LDMC is a measure of leaf tissue density that is related to nutrient residence time within the plant. For LDMC estimation, the fresh mass of leaves weighed with an electronic weighing balance (0.001 g accuracy; Shimadzu, Kyoto, Japan) and dried in the hot-air oven at 80 °C for 48 h to find dry mass values. Values of SLA were expressed in $mm^2 \, mg^{-1}$, while LDMC in $mg \, g^{-1}$.

### 2.4. Leaf Lifespan (LL)

Leaf lifespan (LL) was calculated by the periodic census of tagged leaves [25]. For each species, 150 young, unfolded leaves from five individuals (10 leaves each of three shoots per tree; one sample in Thiruvarur and four in Nagapattinam forest sites) were tagged and observed periodically at a week interval for a period of 24 months (104 weeks) from May 2009 to April 2011. A tagged leaf was considered as dead when completely yellowed or withered from the mother plant. Value of leaf lifespan expressed in months.

### 2.5. CSR Strategies of Trees

The CSR strategies calculator tool (StrateFy) developed by Pierce et al. [26] was utilized to estimate scores for the three principal strategies viz., C-selected "competitor", S-selected "stress-tolerators", and R-selected "ruderal". Data on LA ($mm^2$), SLA ($mm^2 \, mg^{-1}$), and LDMC ($mg \, g^{-1}$) were typed as inputs in preformatted StrateFy tool. The Grime's CSR scheme is useful to compare plant functional strategies across species, populations, communities, ecosystems, and biomes. C-selected species are termed as resource acquisitive,

S-selected are known as resource conservative, while R-selected are stress avoiders and survive in an inactive state as seeds.

### 2.6. Statistical Analyses

Karl Pearson's coefficient of correlation (r) was used to compute the relationship between traits. Partial correlation analysis was applied to test the relationship between a dependent variable and one of the independent variables, by excluding the effect of other variables (kept as constant). To normalize the data, plant trait values were transformed to natural log (ln) prior to the statistical analyses. The significance of the correlation coefficient (r) was verified and confirmed with the Student *t*-test. The significance level $\alpha = 0.05$ was used. STATISTICA 10.0 was used for statistical analyses.

## 3. Results

### 3.1. Specific Leaf Area (SLA)

Specific leaf area was ranged from a minimum of $5.52 \pm 0.20$ mm$^2$ mg$^{-1}$ (*Maytenus emarginata*) to a maximum of $15.782 \pm 0.25$ mm$^2$ mg$^{-1}$ (*Ehretia pubescens*) in TDEFs (as illustrated in Figure 2). SLA varied nearly threefold among species. The mean value of SLA was $10.95 \pm 2.67$ mm$^2$ mg$^{-1}$. Among 44 species, both *Ehretia pubescens* and *Allophylus serratus* showed highest SLA values ($15.78$ mm$^2$ mg$^{-1}$), followed by *Lannea coromandelica* ($15.71 \pm 0.29$ mm$^2$ mg$^{-1}$) and *Aglaia elaegnoidea* ($14.49 \pm 0.31$ mm$^2$ mg$^{-1}$), while the lowest SLA value was recorded for *Maytenus emarginata* ($5.52 \pm 0.20$ mm$^2$ mg$^{-1}$), followed by *Garcinia spicata* ($6.9 \pm 0.11$ mm$^2$ mg$^{-1}$) and *Manilkara hexandra* ($7.21 \pm 0.21$ mm$^2$ mg$^{-1}$). Coefficient of variation was 24.38%.

### 3.2. Leaf Dry Matter Content (LDMC)

Each species had $434.41 \pm 83.16$ mg g$^{-1}$ LDMC (as illustrated in Figure 2) in study area. LDMC ranged from $302.9 \pm 0.32$ (*Anacardium occidentale*) to $605.72 \pm 3.02$ mg g$^{-1}$ (*Pterospermum canescens*). LDMC varied twofold among species. Coefficient of variation was 19.14%. Among species, *Pterospermum canescens* showed the highest LDMC value ($605.72 \pm 3.02$ mg g$^{-1}$) followed by *Diospyros ferrea* ($593.05 \pm 3.4$ mg g$^{-1}$) and *Drypetes sepiaria* ($574.04 \pm 2.09$ mg g$^{-1}$), whereas a low LDMC value obtained for *Anacardium occidentale* ($302.9 \pm 0.32$ mg g$^{-1}$), *Benkara malabarica* ($315.6 \pm 4.81$ mg g$^{-1}$) and *Catunaregam spinosa* ($324.2 \pm 2.61$ mg g$^{-1}$).

### 3.3. Leaf Lifespan (LL)

The mean LL of trees in study area was $12.72 \pm 4.28$ months (as illustrated in Figure 2). Coefficient of variation recorded as 5.44%. *Garcinia spicata* had the highest LL ($23.38 \pm 0.29$ months), while *Phyllanthus reticulatus* ($7.93 \pm 0.37$ months), *Catunaregam spinosa* (8.48 months), *Securenega leucopyrus* (8.13 months), and *Pterospermum canescens* (9.38 months) showed lesser LL.

### 3.4. CSR Strategies of Trees

Trees with S strategy dominated TDEFs. The mean score of C, S, and R strategies were 29.09%, 65.26%, and 5.56%, respectively. One-third of all species had S/CS (15 species, 34.09%) strategy, followed by CS (14 species, 31.82%), S/SR (six), and S (five), whereas single species each showed C/CS and S/CSR.

### 3.5. Relationship between Leaf Traits

Pearson's coefficient of correlation showed a statistically significant negative relationship between SLA and LDMC (r = $-0.74$, $p < 0.01$; $t_{42}$, $n = 44$, $p < 0.001$), SLA and LL (r = $-0.57$, $p < 0.01$; $t_{42}$, $n = 44$, $p < 0.001$), whereas a significant positive relationship existed between LDMC and LL (r = 0.38, $p < 0.02$; $t_{42}$, $n = 44$, $p < 0.01$). From these relationship values, it was inferred that an increase in SLA decreased LDMC and LL, while an increase in LL increased LDMC and decreased SLA in the study area.

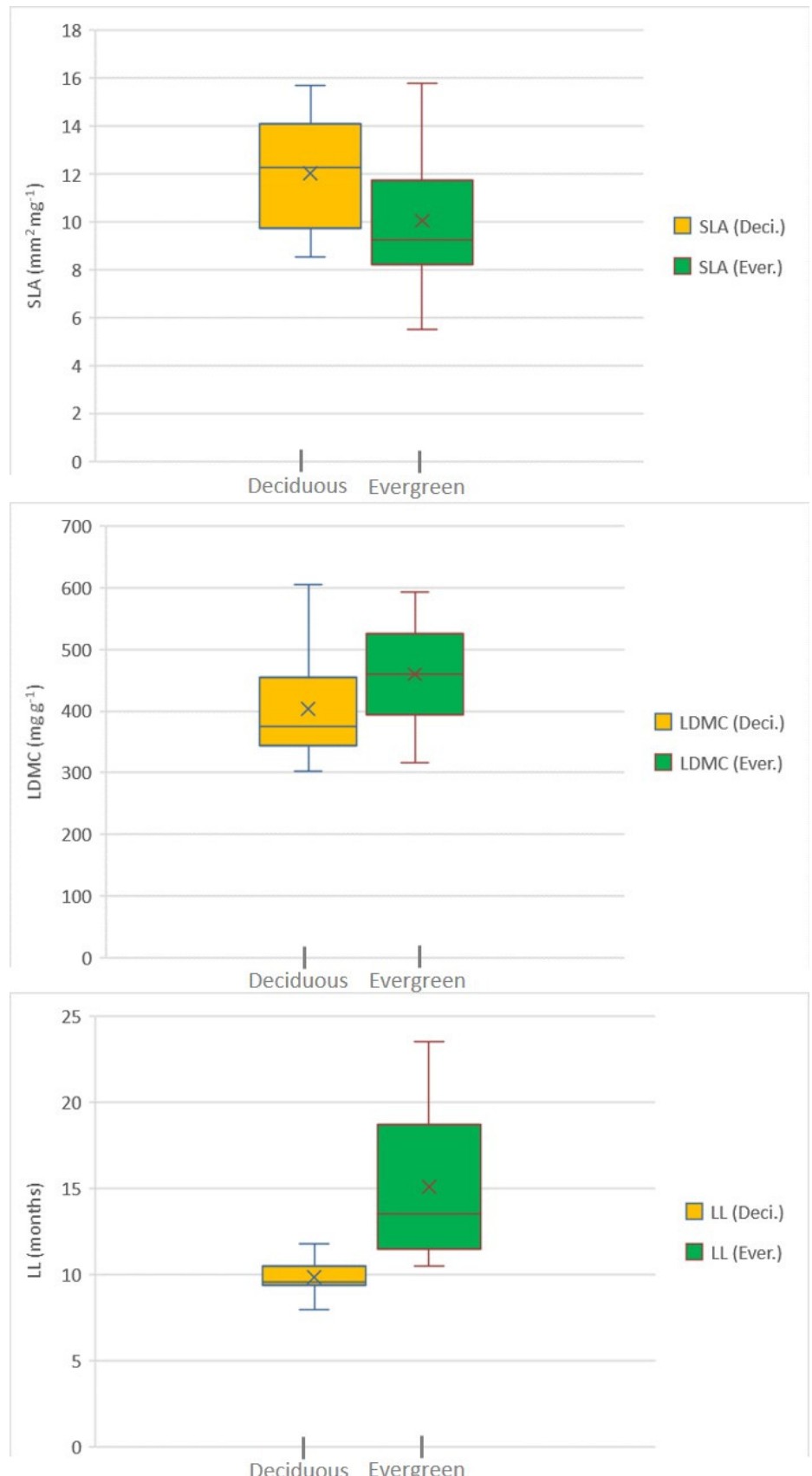

**Figure 2.** SLA, LDMC, and LL of evergreen and deciduous trees in tropical dry evergreen forests of peninsular India.

While verifying the significance of the correlation coefficient of leaf traits with '*t*' test, all the values (r) showed significance. The *t* values differed among leaf traits relationship. The *t* value of SLA and LDMC relationship was 7.13 ($p < 0.001$), SLA and LL was 4.50 ($p < 0.001$), and LDMC and LL was 2.66 ($p < 0.01$).

There was a significant difference between mean trait values (SLA, LDMC, and LL) of evergreens and deciduous species. The mean value of SLA of evergreens notably varied from deciduous (*t* test: $t_{42} = 2.013$; $p < 0.05$). LDMC (*t* test: $t_{42} = 1.98$; $p < 0.05$) and LL (*t* test: $t_{42} = 5.23$; $p < 0.001$) also deviated significantly. Compared to that of deciduous species, evergreens had less SLA, but high LDMC and LL.

### 3.6. Leaf Traits of Evergreen Species

While calculating leaf traits of evergreens alone, the value of SLA was ranged from $5.52 \pm 0.20$ to $15.78 \pm 0.25$ mm$^2$ mg$^{-1}$. Among evergreens, *Allophylus serratus* and *Ehretia pubescens* had the highest SLA value ($15.78 \pm 0.38$; $15.78 \pm 0.25$ mm$^2$ mg$^{-1}$) followed by *Aglaia elaegnoidea* ($14.59 \pm 0.31$ mm$^2$ mg$^{-1}$) and *Canthium coromandelicum* ($13.32 \pm 0.25$ mm$^2$ mg$^{-1}$), whereas the least SLA obtained for *Maytenus emarginata* ($5.52 \pm 0.20$ mm$^2$ mg$^{-1}$), and *Garcinia spicata* ($6.9 \pm 0.11$ mm$^2$ mg$^{-1}$). The mean SLA of evergreens was $10.05 \pm 2.77$ mm$^2$ mg$^{-1}$ (as illustrated in Figure 2).

LDMC ranged from $593.05 \pm 3.4$ mg g$^{-1}$ to $315.6 \pm 4.81$ mg g$^{-1}$ for evergreens (mean $460.35 \pm 79.42$ mg g$^{-1}$). *Diospyros ferrea* had a high LDMC ($593.5 \pm 3.4$ mg g$^{-1}$) followed by *Drypetes sepiaria* ($574.04 \pm 2.09$ mg g$^{-1}$) and *Manilkara hexandra* ($573.57 \pm 1.91$ mg g$^{-1}$), while *Benkara malabarica*, *Canthium coromandelicum* and *Aglaia elaegnoidea* had least values, $315.6 \pm 4.81$, $333.9 \pm 2.96$ and $349 \pm 7.06$ mg g$^{-1}$, respectively (as illustrated in Figure 2).

As to the LL of evergreens, *Atalantia monophylla* had the highest value ($23.53 \pm 0.22$ months) followed by *Garcinia spicata* ($23.38 \pm 0.29$ months) and *Memecylon umbellatum* ($22.48 \pm 0.25$ months), while *Benkara malabarica* ($10.2 \pm 0.20$ months), *Ficus hispida* ($10.8 \pm 0.44$ months) and *Maytenus emarginata* ($11.13 \pm 0.36$ months) secured low values. The mean LL of evergreens recorded as $15.76 \pm 4.50$ months (as illustrated in Figure 2).

### 3.7. Leaf Traits of Deciduous Species

SLA of deciduous species ranged between $8.53 \pm 0.17$ mm$^2$ mg$^{-1}$ (*Syzygium cumini*) and $15.71 \pm 0.29$ mm$^2$ mg$^{-1}$ (*Lannea coromandelica*). Among deciduous species, *Lannea coromandelica* had the highest SLA ($15.71 \pm 0.29$ mm$^2$ mg$^{-1}$) followed by *Breynia vitis-idaea* ($14.57 \pm 0.16$ mm$^2$ mg$^{-1}$) and *Madhuca longifolia* ($14.57 \pm 0.36$ mm$^2$ mg$^{-1}$), while least SLA recorded for *Syzygium cumini* ($8.53 \pm 0.17$), *Ochna serrata* ($9.51 \pm 0.18$), and *Ficus benghalensis* ($9.62 \pm 0.20$). The mean SLA was $12.03 \pm 2.16$ mm$^2$ mg$^{-1}$, (as illustrated in Figure 2).

Of 20 deciduous species, *Pterospermum canescens* ($605.72 \pm 3.02$ mg g$^{-1}$), *Ficus benghalensis* ($527.1 \pm 4.33$ mg g$^{-1}$), *Syzygium cumini* ($474.17 \pm 1.93$ mg g$^{-1}$) and *Ochna serrata* ($464.08 \pm 2.04$ mg g$^{-1}$) had high LDMC, whereas *Anacardium occidentale* ($302.9 \pm 0.32$ mg g$^{-1}$), *Phyllanthus reticulatus* ($323.46 \pm 1.73$ mg g$^{-1}$) and *Catunaregam spinosa* ($324.2 \pm 2.61$ mg g$^{-1}$) scored less. Mean value of LDMC for deciduous species was $403.29 \pm 78.40$ mm$^2$ g$^{-1}$ (range $302.9 \pm 0.32$ to $605.72 \pm 3.02$ mg g$^{-1}$; as illustrated in Figure 2)

Each deciduous species had $9.85 \pm 1.03$ months of LL in the study area (range $7.93 \pm 0.37$ to $11.58 \pm 0.12$ months). Among deciduous species, *Tamarindus indica* ($11.58 \pm 0.12$ months), *Sapindus emarginatus* ($11.15 \pm 0.38$ months) and *Pongamia pinnata* ($10.68 \pm 0.33$ months) had high LL, whereas *Phyllanthus reticulatus* ($7.93 \pm 0.37$ months) *Securenega leucopyrus* ($8.13 \pm 0.27$ months) and *Catunaregam spinosa* ($8.48 \pm 0.34$ months) had low LL value in TDEFs (as illustrated in Figure 2).

### 3.8. Relationships among Traits

Two traits, LDMC and LL jointly explained 80% of the variation in SLA ($R_{1.23} = 0.80$ (r$^2$); $p < 0.01$; *df* = 43). The variation explained by LDMC in SLA was 46% ($R_{12.3} = 0.46$ (r$^2$); $p < 0.01$) when considered LL as constant. Similarly, while excluded the effect of LDMC the

variation explained by LL in SLA was 22% ($R_{13.2} = 0.22$ ($r^2$); $p < 0.01$). These results signify that SLA had a closer relationship with LDMC than LL. In other words, the LDMC had a closer relationship with SLA than LL. Simple correlation also proved this fact.

The SLA and LL together explained 74% of variation in LDMC ($R_{2.13} = 0.74$ ($r^2$); $p < 0.01$; *df* = 43). While excluded the effect of LL (kept as constant) the variation explained by SLA in LDMC was 46% ($R_{12.3} = 0.46$ ($r^2$); $p < 0.01$; *df* = 43). Likewise, when kept SLA as constant the variation explained by LL in LDMC was just 7% ($R_{23.1} = 0.07$ ($r^2$); $p > 0.05$; *df* = 43). These results indicate that LDMC had closer link with SLA than LL.

In cooperation, SLA and LDMC expounded 56% of variation in LL ($R_{1.23} = 0.56$ ($r^2$); $p < 0.01$; *df* = 43). When kept SLA as constant LDMC explained just 7% of variation in LL ($R_{23.1} = 0.07$ ($r^2$); $p > 0.05$; *df* = 43). Similarly, while kept LDMC as constant the SLA expounded 22% of variation in LL ($R_{13.2} = 0.22$ ($r^2$); $p < 0.01$). These findings point out that the LL had a closer relationship with SLA than LDMC.

LDMC explained relatively more proportion of variation in SLA (55%) than LL (32%); hence, the relationship between SLA and LDMC was stronger than the relationship involving SLA and LL. Likewise, the LL expounded 32% of the variation in SLA, while explained just 14% in LDMC; thus, the correlation between SLA and LL was stronger than the relationship between LL and LDMC.

## 4. Discussion

### 4.1. Core TDEF Tree Species, Physiognomy and Leaf Types

Blanchflower [21] designated a list of 18 species as core TDEF species. The current study included most of the core tree species and other common species of TDEF. The mean annual temperature and rainfall of Coromandel Coast districts not varied considerably, and thus, the current study area can be considered as representatives of extant TDEFs in Tamil Nadu.

The dominance of evergreen leaf habit (84.22% of individuals) in poor-nutrient habitat is in concurrence with the earlier report of Givnish [27], who found an abundance of evergreen leaf habit in nutrient-poor soils. Deciduous trees are less common compared to that of evergreens in terms of species richness as well as stand density; this characteristic provides ever-greenness to TDEFs throughout the year. Similarly, compound-leaved species are less common than simple-leaved ones. TDEFs occur in low elevation and semi-arid environment, but with combination of low-nutrient soil, high temperature and longer dry season these forests were flourishing as a unique evergreen forest ecosystem.

### 4.2. Specific Leaf Area

The SLA plays a central role in various plant strategy schemes [2,28,29], where it considered as an index of a species' position along the continuum between a high SLA that realizes a fast resource acquisition and growth, and a low SLA that realizes a high resource conservation and persistence. The mean ($10.95 \pm 2.67$ mm$^2$ mg$^{-1}$) and range of SLA ($5.52 \pm 0.20$ to $15.78 \pm 0.25$ mm$^2$ mg$^{-1}$) recorded in TDEF are comparable and well within the range found in forest types around the world (as illustrated in Table 2). Observations on SLA (mean and range) suggest that species of dry, dry deciduous and evergreen forests are associated with low SLA leaves (range 3.87–26.68 mm$^2$ mg$^{-1}$) (as illustrated in Table 2).

**Table 2.** Leaf trait values recorded from forest types around the world (ecological context of studies provided in parenthesis).

| Trait and Forest | Value | Reference |
|---|---|---|
| **SLA mm$^2$ mg$^{-1}$ (Mean and Range)** | | |
| Mixed evergreen-deciduous oak forest, southern France (species ranking based on functional traits) | 6–31.7 (range)<br>13.4 ± 1.2 (mean) | [1] |
| Evergreen shrublands, France (functional traits and plant growth) | 9.91–26.37 (range) | [10] |
| Subtropical broad-leaved forests, South Carolina, USA (Generality of leaf traits relationship) | 4.21–9.79 (range) | [17] |
| Dry forests, Australia (leaf area in relation to leaf size) | 2.7–23.3 (range) | [30] |
| Old-growth lowland tropical forest, Central Panama (leaf functional traits and growth form) | 3.87–26.68 (range) | [31] |
| Multiple source data set (leaf dark respiration and functional traits) | 7.1 (shrubby-evergreen) (mean)<br>14.0 (shrubby-deciduous) (mean)<br>13.7 (broad-leaved deciduous) (mean)<br>8.9 (broad-leaved evergreens) (mean) | [32] |
| Dry forest, Panama (leaf traits of canopy trees and precipitation gradients) | 8.5 (mean) | [33] |
| Dry forest, Costa Rica (plant functional types in tropical dry forests) | 10.2 (mean) | [34] |
| Dry forest, Panama (photosynthetic characteristics and associated traits) | 12.6 (mean) | [35] |
| Dry forest, Venezuela (cost-benefit relationships in deciduous and evergreen leaves) | 18.6 (mean) | [36] |
| Dry deciduous forest, Australia (cost-benefit analysis of trees) | 9.9 (mean) | [37] |
| Dry deciduous forest, India (species and site effects on leaf traits) | 11.4 (mean) | [38] |
| Deciduous forest, Central Western Argentina (leaf traits as indicators of resource-use strategy) | 11.2 (mean) | [39] |
| Chaparral forest, California (leaf traits and resprouting ability) | 3.52 to 17.5 (range) | [40] |
| Subtropical monsoon forest, China (leaf morphology and functional ecophysiological traits) | 13.56 (Evergreen) (mean)<br>10.86 (Deciduous) (mean) | [41] |
| Various forest types (estimation of laminar leaf thickness by SLA and LDMC) | 8.3 (mean) | [42] |
| Global forests (leaf traits, climate, and soil) | 2–41 (range) | [43] |
| Tropical forests, Bolivia, Brazil, and Costa Rica (functional trait diversity and aboveground biomass productivity) | 12.63 (mean)<br>9.84–17.1 (range) | [44] |
| Various forest types, China (included herbs; prediction of soil fertility using SLA and LDMC) | 1.89–94.99 (range) | [45] |
| Global database (all life forms; biomass allocation and plant functional traits) | 2.3–122.8 | [46] |
| Semiarid forest (biomass allocation and plant functional traits) | 8.62–21.86 | [46] |
| Temperate forest, India (plant functional diversity and carbon accumulation) | 6.95–13.02 (range)<br>9.27 (mean) | [47] |
| Dry deciduous forest, India (foliar demand and resource economy of nutrients) | 10.9 (mean) | [48] |
| Deciduous mixed forest (impacts of mining and quarrying on functional traits) | 26.14 (mean) | [49] |
| Various forest types (global convergence in plant functioning) | 15.0 (mean) | [50] |
| **LDMC mg g$^{-1}$ (mean and range)** | | |
| Mixed evergreen-deciduous oak forest, South France (species ranking based on functional traits) | 338 ± 16 (mean)<br>131–472 (range) | [1] |
| Subtropical forests, South Carolina, USA (functional traits and plant growth) | 124.58 to 557.38 (range) | [10] |
| Mature Mediterranean maquia vegetation, Eastern Iberian Peninsula, Spain (leaf traits and resprouting ability) | 412.3 ± 71.7 (sprouters)<br>415.9 ± 31.1 (non-resprouters)<br>441.3 ± 57.4 (tertiary period species)<br>383.2 ± 57.5 (quaternary period species) | [40] |
| Various forest types (estimation of laminar leaf thickness by SLA and LDMC) | 414.7 (mean) | [42] |
| Global database (included herbs; prediction of soil fertility using SLA and LDMC) | 48–517.8 (range) | [45] |
| Temperate semiarid forest (all life forms; biomass allocation and plant functional traits) | 260–510 (range) | [46] |
| Dry deciduous forest, Bolivia (hydraulics and life history of tropical dry forest trees) | 322 ± 93 (mean)<br>230–480 (range) | [51] |
| Deciduous forest, central-western Argentina (measurement methods of LDMC) | 359 ± 36 (mean)<br>134 to 474 (range) | [52] |
| Various forest types, China (latitudinal variations and leaf-morphological traits) | 44.46–775.7 (range) | [53] |
| Deciduous mixed forest (leaf traits and herbivory on deciduous and evergreen trees) | 233.3 (mean) | [54] |
| **LL months (mean and range)** | | |
| Subtropical forest, South Carolina, USA (functional traits and plant growth) | 7.5–19.5 (range) | [10] |
| Dry forests, New Mexico (generality of leaf trait relationships) | 11.1 (mean) | [17] |
| Temperate deciduous and tropical evergreen forests, Mexico (leaf demography and leaf traits) | 8.3 ± 0.08 (Deciduous) (mean)<br>15.0 ± 0.48 (Evergreen) (mean) | [18] |
| Old-growth lowland tropical forest, Caribbean coast, Central Panama (leaf traits of canopy trees and precipitation gradients) | 5.8–23 (range) | [33] |
| Dry forest, Venezuela (leaf traits as indicators of resource-use strategy) | 8.4 ± 1.3 (mean) | [39] |
| Temperate forest, India (foliar demand and resource economy of nutrients) | 17.96 ± 3.41 (mean)<br>12.71–29.76 (range) | [48] |

**Table 2.** *Cont.*

| Trait and Forest | Value | Reference |
|---|---|---|
| **SLA mm² mg⁻¹ (Mean and Range)** | | |
| Tropical dry forest, Brazil (leaf traits and herbivory on deciduous and evergreen trees) | 11.75 (evergreen) (mean)<br>6.0 (deciduos) (mean) | [54] |
| Aseasonal tropical forest, New Guinea (mineral cycling and plant traits) | 16.48 (mean) | [55] |
| Great Smoky Mountain, south-eastern North America | 3.9–6 (range) | [56] |
| Broad-leaved evergreen forest, Japan (leaf dynamics and shoot phenology) | 18–51.6 (range) | [57] |
| Tropical forest, Costa Rica (longevity of leaves) | 24 (mean) | [58] |
| Broad-leaved forest, Japan (leaf survival of woody plants) | 2.7–5.33 (mean) | [59] |
| Dry forest, Australia (leaf attributes across four different habitats in Australia) | 6.3 ± 0.5 (mean) | [60] |
| Subtropical forest, China (leaf economics of evergreen and deciduous tree species) | 17.1–30.9 (evergreen)<br>5.3–8.3 (deciduous) | [61] |

Evergreen species with low SLA (small and thick leaves) dominated TDEFs in terms of species richness and density. Existence of low SLA trees in TDEFs could be due to low-nutrient soil (C and N concentrations), higher mean annual temperature (~33 °C), and longer dry period (5–6 months). The relationships between longer dry period and low SLA, and low SLA and oligotrophic habitat were reported consistently around the world, e.g., [30,31,62,63].

### 4.3. Leaf Dry Matter Content

The range and mean LDMC of TDEF are comparable with that of studies concentrated on various plant functional traits in tropical and temperate vegetation around the world (as illustrated in Table 2).

The occurrence of good proportion of evergreens, longer dry period, and poor concentrations of soil-nutrients could were contributed to a relatively higher mean LDMC (434.40 mg g⁻¹) of trees in TDEFs. The need for higher LDMC under dry, low-nutrient environment was emphasized by many scientists who researched on a range of forests around the world. However, the present study concentrated on just 44 species from TDEFs, so more studies with large number of species are essential for general inferences.

### 4.4. Leaf Lifespan

Leaf lifespan of TDEF trees is in line with available pieces of literatures that recorded leaf lifespan of trees in a range of forest types around the world (as illustrated in Table 2). Leaf lifespan of deciduous species in TDEF (9.85 ± 1.03 months) is higher than the deciduous forest of Australia (6.3 ± 0.5 months; [60]) and dry forest of Venezuela (8.4 ± 1.3 months; [36]. The LL must be longer in dry forests, like TDEFs, because trees are flourishing on low-nutrient soils, dominated by evergreens both in terms of species richness and density and experiencing 5–6 dry months in a year. In general, habitats with poor concentrations of essential soil nutrients associate with a relatively higher LL e.g., [64]. On the nutrient-limited habitat, the leaves need longer duration to pay back their construction cost than leaves of nutrient-rich habitats [65,66].

TDEFs have both the leaf habits, deciduous and evergreens. Normally, forests flourishing on water-limited habitats have both deciduous and evergreen species e.g., [17]. The differences in LL between evergreen and deciduous species could be useful to partitioning the available resources and reduce the competition among species who compete for the available nutrients [65].

### 4.5. CSR Strategies of Trees

The StrateFy tool of Pierce et al. [26] found tropical and subtropical broadleaf forests with CS/CSR strategies. However, the number of species used to calculate strategies in Pierce et al. [26] is 11-fold higher than in that of the present study. The CS selection is linked with a warm and lesser temperature seasonality. The S strategy score of TDEF trees (65.26) indicated a relatively higher biomass allocation to persistent tissues. A longer longevity of tissues was considered as advantageous under the low-nutrient habitat. Species of S

strategy was considered as stress tolerant and resource conservators' viz., associated with a low SLA, high LDMC and LL. TDEFs are rich in evergreens, and hence, a significant proportion tree community showed the S strategy. C selected competitors tended to associate with maximum resource acquisition. In general, they occupy habitats with higher productivity and lesser disturbance and allocate a higher proportion of biomass to vegetative growth, thus producing a high SLA, low LDMC and LL. In general, deciduous species tends to associate with a higher C values. Existing TDEF sites are protected by local people on religious grounds and therefore experience little disturbance. The mean CSR scores of common TDEF species (29.69:64.09:6.22) did not vary from core species (28.95:67.45:3.60). This observation demonstrated that those species (both core and common TDEF species) adapted for drier environment and low-nutrient are flourishing in TDEFs.

Evergreen species dominate the study area (84.22%) with a high evergreen-deciduous ratio of 5.34: 1. Evergreen species (generally associated with low SLA, high LDMC and LL) have a number of advantages in nutrient-poor habitat. As mentioned earlier, TDEF is one of the poor nutrient habitats, flourishing on coastal sandy alluvium. Water holding capacity, nitrogen (N, range, 1.01–1.12 mg g$^{-1}$), and phosphorus (P, range, 31.4–65.2 µg g$^{-1}$) concentrations are low. The soil C/N ratio is high (range, 34.04–38.24) [67]. It is evident from the present study that evergreens are abundant, thus, TDEF is an old-growth one, not an early successional forest. Earlier, Dewalt et al. [68] recognized sites of TDEF as old-growth forests (largely undisturbed for at least 71 years).

*4.6. Relationship between Traits*

The relationships of leaf traits obtained in the present study are generally similar, as found in other forest sites from local, regional, and global scales (e.g., [69]). In general, leaves of deciduous (low SLA leaves) are associated with higher rates of photosynthesis [19] and relative growth rates [70]. It was found that the decidous trees are fast nutrient users, they generally have a high SLA and LL and a low SLA, while the opposite is true regarding evergreens in which nutrient conservation is important (e.g., [71]). The community and ecosystem weighted mean of traits largely control the productivity and biogeochemical cycling (e.g., [72]).

Specific leaf area and leaf dry matter content reflects a fundamental trade-off in plant functioning between rapid (high SLA, and low LDMC species) and slow production (low SLA and high LDMC species) of biomass [71]. Similarly, such a relationship between SLA and LDMC, as observed in the present study, concurs with earlier reports, e.g., [1,72] (as illustrated in Table 3). Garnier et al. [1] ranked species according to SLA and LDMC and found no substantial change temporally or across different environments. The study emphasized the relationship between these traits is quite general across species, ecosystems, and biomes. Wilson et al. [7] addressed the resource-use strategy of species with a combination of SLA and LDMC. They recognized the LDMC as a best single variable in locating plant species on a resource use axis. Generally, leaves of low SLA have a higher amount of biomass than water in per unit area compared to leaves of high SLA. In this study, leaves have 62% of biomass and 38% of water per unit area. This phenomenon largely generates negative correlation between SLA and LDMC.

**Table 3.** Correlation between leaf traits recorded from forest types around world.

| Trait Pair | Relationship (r or r$^2$) | Reference |
|---|---|---|
| SLA-LDMC, Mixed evergreen-deciduous oak forest, south France (species ranking based on functional traits) | −0.73 | [1] |
| SLA-LDMC, British flora (SLA and LDMC as alternative predictors of plant strategies) | 0.30 (negative) | [7] |
| SLA-LDMC, Evergreen shrub land, south France (species ranking based on leaf traits) | −0.81 | [10] |
| SLA-LL, Diverse forests (LEAVES dataset) (leaf lifespan in relation to leaf, plant, and stand characteristics) | 0.54 (negative) | [12] |
| SLA-LL. Global forests (generality of leaf trait relationships) | −0.9 ($p < 0.0001$) | [17] |
| SLA-LL, Montane cloud forest, Mexico (leaf demography and leaf traits) | −0.85 ($p < 0.0001$) | [18] |
| SLA-LL, Tropical lowland forest, Panama (leaf functional traits of growth forms) | 0.50 (negative) | [31] |
| SLA-LL, various forests (worldwide leaf economics spectrum) | 0.52 (negative) | [69] |
| SLA-LDMC, Sand dunes, China (SLA and LDMC of dynamic ecosystem) | 0.196–0.454 (negative) | [73] |
| LDMC-LL, Estonian forest (components of leaf dry mass, thickness and density alter leaf photosynthesis) | Positive | [74] |
| SLA-LL, Tropical forest, Amazon (leaf lifespan as a determinant of leaf structure and function) | −0.66 (negative) | [75] |
| SLA-LL, Typical Hapludalf, southwestern Wisconsin (effect of leaf longevity of biomass production) | 0.92 (negative) | [76] |
| SLA-LL, Alpine forest, Japan (intraspecific variations of leaf traits) | −0.68, −0.73, −0.87 | [77] |
| LDMC-LL, Temperate ecosystem, France (ranking of species using leaf traits) | Positive | [78] |
| SLA-LDMC, Temperate forest, Italy (plant adaptive responses during primary succession) | −0.357 | [79] |
| LDMC-LL, Tropical savanna, Africa (leaf traits of woody species across climate and soil-fertility gradients) | Positive | [80] |
| SLA-LDMC, Global forests (leaf trait covariation across climates and biomes) | −0.53 | [81] |
| SLA-LDMC, Temperate forest, China (leaf trait variation and correlation across biological and spatial scales) | −0.21 ($p < 0.001$) | [82] |

The correlation in the relationship between SLA and LL (r = −0.57; $p < 0.01$) in TDEFs is in agreement with earlier reports (as illustrated in Table 3). A negative relationship between SLA and LL was reported by Westoby et al. [83]. With 73 Australian species, Wright and Westoby [84] recorded a negative correlation between SLA and LL. With a large sample size (*n* = 678), Wright et al. [13] found a negative relationship between SLA and LL (*n* = 678). In a field-based inventory on a poor-phosphorus Australian soil, Wright et al. [69] showed a negative relationship between SLA and LL (r = −0.72, *n* = 128).

In our study area, evergreens had a higher mean LL (15.11 ± 4.50 months) than in that of deciduous (9.85 ± 1.03 months; (*t*-test: $t_{42}$ = 5.23; $p < 0.001$). Evergreens (low SLA) are widely associated with higher LL than in that of deciduous [12,74,85,86]. Likewise, low SLA species were shown to achieve greater LL in a variety of habitats [70,87]. Moreover, on the global scale, leaf traits such as SLA and LL have a correlation with one another [88–90]. In general, species of low SLA widely associated with high LL, while high SLA linked with low LL. This characteristic widely produces a negative link between SLA and LL in the lab as well as field-based leaf trait studies.

In this study, evergreens have a low SLA (10.05 ± 2.77 mm$^2$ mg$^{-1}$) than in deciduous (12.03 ± 2.16 mm$^2$ mg$^{-1}$). This generality, evergreen-low SLA, and deciduous-high SLA was reported in field as well as laboratory-based plant functional trait studies [19,36]. Usually, high-rainfall forests have a larger proportion of higher SLA species than in low-rainfall habitats [89,90]. The current study area receives relatively poor rainfall compared to that of wet forests; hence, they support greater numbers of low SLA species.

A positive relationship between LDMC and LL was recorded in this study. Earlier, many researchers found a strong link between LDMC and LL. For example, Niinements [74] reported a strong relationship between LDMC and LL with 44 tree species from an Estonian forest (see Table 3 for previous studies). Studies concentrated on the relationship LDMC-LL are limited compared to other traits. Usually, evergreens have a high LDMC leaves. Generally, evergreen leaves live longer than deciduous (low LDMC), thus, largely LDMC is positively correlated with LL.

### 4.7. Habitat Fertility and Leaf Traits

Leaf traits are tightly linked with habitat fertility. Soil nutrient concentrations recorded in study area show that TDEFs are nutrient-poor forests, having very poor N and P concentration [20,91,92]. Nutrient poor habitat also has advantages. Especially, poor N fertility may check the invasion of highly competitive plant species that need high N availability for their survival [93]. Scores of literatures showed the relationship between soil nutrient and physiognomy e.g., [26,94,95].

The nutrient-poor soil of our study area contributes to low SLA, high LDMC and LL. Result obtained in this study is comparable with those of several literatures; they consistently reported that soil nutrition, especially low-P habitat, favor smaller leaf size [96]. Previously, nutrient availability of habitat was believed to be important predictors in the determination of plant trait values [65].

The relationship between leaf traits and soil fertility was widely reported in the literature. Effects of habitat fertility on LL were reported by observational and experimental studies. For example, fertilization decreased LL of *Larrea tridendata* [97]. Similarly, low SLA leaves are often associated with poor-N [98] as well as poor P soil [99].

The study area endowed with large number of evergreens. By and large, leaves of evergreens have lower N concentration; therefore, decomposition rate is lesser for N poor evergreen leaves than N rich deciduous leaves [100]. A positive relationship often prevails between soil nutrition and mean leaf lifespan in forest ecosystem. Thus, oligotrophic environment as forest under study usually have a greater quantity of evergreens and produce nutrient-poor litters.

### 4.8. Traits of Evergreen and Deciduous Trees

Leaf trait values of deciduous varied from evergreens in TDEFs. Earlier, a number of studies on leaf functional traits recorded a contrasted values for leaf habits (evergreen and deciduous). For example, the mean SLA of deciduous (13.56 mm$^2$ mg$^{-1}$) significantly varied from evergreen (10.86 mm$^2$ mg$^{-1}$) [41]. The average LL of evergreen (15 ± 0.08 months) differed notably from deciduous (8.25 ± 0.48 months) [18], while the mean LDMC of evergreen was considerably deviated from deciduous e.g., ([53,100]), (as illustrated in Table 3)

### 5. Conclusions

This study sheds light on important leaf traits of tropical dry evergreen forests of southern Coromandel Coast, Peninsular India. The relationships observed between leaf traits in our study is common in tropical and temperate broadleaf forests, and thus, the present study could contribute a notable supplement to research on plant functional traits, especially leaf traits at a larger scale. The present study supports a widely reported important leaf traits relationships, namely, SLA and LL, SLA and LDMC, and LDMC and LL. The S strategy score indicates trees of TDEFs allocating a relatively higher biomass to persistent tissues thereby surviving on low-nutrient habitat. Trees of TDEFs experience less disturbance and survive on the oligotrophic habitat; thus, they are associated with a low SLA, high LDMC and LL. If more plant functional trait studies are conducted, we could learn the complete ecophysiology of an understudied, unique, and endangered forest type (TDEF) in the Indian subcontinent.

**Author Contributions:** Conceptualization, M.U.; methodology M.U. and T.S.; validation, T.S.; investigation, M.U.; resources, M.U.; data curation, M.U. and T.S.; writing—original draft preparation, M.U.; writing—review and editing, M.U.; visualization, M.U.; supervision, T.S.; project administration, T.S.; funding acquisition, M.U. All authors have read and agreed to the published version of the manuscript.

**Funding:** This research was funded by Department of Science and Technology, Govt. of India, New Delhi grant number DST/INSPIRE Fellowship/2009/[xxxix] 17 March 2010 (IF No. 10051).

**Institutional Review Board Statement:** Not applicable.

**Informed Consent Statement:** Not applicable.

**Data Availability Statement:** Data provided within the manuscript.

**Acknowledgments:** The first author, M.U., thanks K. Ajithadoss, formerly Professor and Head, Department of Botany, Presidency College (Autonomous), Chennai, for his help and support through all means since 2002.

**Conflicts of Interest:** The authors declare no conflict of interest. The funders had no role in the design of the study; in the collection, analyses, or interpretation of data; in the writing of the manuscript, or in the decision to publish the results.

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
