# Peer review of "Leaf Traits of Trees in Tropical Dry Evergreen Forests of Peninsular India"

_2673-4133, doi:10.3390/ecologies2030015_

Round 1

Reviewer 1 Report

The manuscrit ‘Leaf traits of trees in tropical dry evergreen forests of peninsular India’ by Udayakumar & Sekar, refers to a good work of measuring and compiling relevant plant functional traits, with the result of a complete data set of 44 tree species by three leaf functional traits. This is a very promising approach to elucidate key processes and main functioning of the tree community studied, and therefore of the whole ecosystem, from the points of view of biomass and C balance, biogeochemical cycles, conservation of soil, the plant functions and diversity, and even restoration ecology. The ecosystem studied is of great interest, and indeed less known; and the data were obtained through standard consistent methods.

This potential development, however, is not optimally performed in the paper. Instead, after presenting the descriptive results (not in the best way, to my view), the discussion emphasizes the own results, compares them with other literature results on plant functional traits (from similar or very dissimilar ecosystems) and also include comments trying to link the patterns observed in the distribution of plant functional traits  among the species studied with ecological functioning and even with ecosystem conservation or restoration, but merely through informal referencing, which may lead to speculative comments.

As for the different parts of the manuscript:

  1. The abstract could be shortened, possibly reducing the detail given the meaning of well-known plant functional traits, the numerical findings, etc.
  2. In the Material and Methods section, there would be useful including some quantification of the relevance of the species studied in plots studied or in the ecosystems, (cover, abundance, frequency, ecosystem function), given the comments put in the discussion on the relevance of evergreens vs. deciduous, etc.
  3. The Results part includes good comments on the values obtained, but there is some redundancy in some cases. Moreover, some graphic presentation of the main results would be desirable, as complement of the raw table of data.
  4. The discussion is, as commented, not very suggestive. It is expected from this section to present different approaches of the study subject, than that presented in the Results, and to do this in a more consistent way than here. Comments on the nutrient poorness of the soils, on drought limitation, on secondary succession, et al., should be more consistently worked. Moreover this discussion is really too long (in contrast to its own limitations), which is due to redundancy in many parts.
  5. Conclusions are not properly so; they have to derive from own results and discussion.
  6. The list of references seems too long, for the study done.
  7. In addition, English writing (and formal presentation of the manuscript) is not always good. Especially the discussion is rich in grammatical inconsistencies or mistakes (but I’m not English-native).

Reviewer 2 Report

Thanks to the authors for giving me the opportunity to review their paper.

Although the study is interesting and provides a lot of information on the subject, there are some aspects that can be improved:

Changing the abstract by eliminating the abundance of numerical data, data that can be presented in the body of the paper;

Adding information in Chapter 2 with more research from studies conducted by other researchers;

Addition in Chapter 3 and 4 of details on the data collection data processed

Reviewer 3 Report

My comments are below.

Abstract

Authors’ wrote: "The relationships found between SLA and LDMC (negative); LDMC and LL (positive); and, SLA and LL (negative) were in concurrence with earlier findings."

So please, briefly describe the novelty of this study. 

Introduction

Authors’ wrote: "Assessing relationships among leaf traits and how traits differ between or among growth forms (herb, shrub, tree and liana) and plant functional types (deciduous, evergreen) can give important insights
into the selective pressures that have shaped the evolution of the
world’s vegetation"

Please, expand the issue of the "insight, that can be given into tne selective pressures that have shaped the evolution of the world’s vegetation". 

Authors’ wrote: "The degree of relationships between important leaf traits was estimated to find whether the broadly reported leaf trait relationships namely, SLA-LDMC, SLA-LL, and LDMC-LL exist in TDEFs or not?"

I suppose, authors should set up a hypothesis based on the current state of knowledge and then tested it. 

IF <The relationships found between SLA and LDMC (negative); LDMC and LL (positive); and, SLA and LL (negative) were in concurrence with earlier findings>, THEN the novelty/originality is uncertain and shall be improved.

2.4. Statistical analyses

The statistical anlayses should be improved. I suggest you to involve the sites as a random variable in the analyses.

The graphical presentation of results is in fact absence. Please, improve this part of the manuscript, cause it significantly help the readers of your manuscript, and therefore, the contribution of your manuscript to the scientific community.

Moreover, the sites are located within a 4 miles (6 km) radius, with a two located a 20 miles (30km) away. That means, you prepared the study in two geographically distinct sites with 2 and 8 study stands within each site respectively. It is not well-planned study design.

Especially that this forest type covers, as you wrote "a sum of 1,08,000 km2". (Twice as much as an area of Sri Lanka.)

As it is known, both soil and climate/latitude drives plant SLA (Gong, Gao 2019 https://doi.org/10.1016/j.gecco.2019.e00696; Gong et al 2020; https://doi.org/10.1016/j.gecco.2020.e00904 ). Therefore I think, that two geographically distinct sites are insufficient for any general conclusions.

In addition, I suppose, that due to the closely located areas of study you obtained a low SD values in the Table 2.

Authors’ wrote: "A total of 100 leaves (10 leaves each from ten individuals) along with petiole were measured)".

It means, you have tested a 1 individual tree per each of 10 study sites?

Please, add the number of individual plants (per plant species) tested in each of 10 study site and the total number of leaves and individual plants tested in each study site.

For example : site1, species Albizia lebbeck, X individual plants and Y leaves from each individual plant respectively (finally : X*Y leaves were taken from site1). 

Then your methods will be more transparent, and the results clarified.

Table 2.
I recommand you to present this results in graphical form. For example as box-plots. Moreover, I suggest you to involve the taxonimic identity of plant species (family, order) and regroup the tested plant taxa according to their taxonomy. If you marked both evergreen and decidious plants by two distinct colours, the figures will be easier to read.

DISCUSSION

Authors’ wrote: "Observations made from the present investigation are in accord with the findings from similar studies carried out elsewhere by different investigators."

Please, clarify the results of other studies of SLA, LL and LDMC in evergreen vs decidious plants, and discuss the contribution of this manuscript to the entire scientific community.

Authors’ wrote (in Abstract): This study adds baseline data on key leaf traits to plant functional trait database of India.

I think, it is a merit of this study. Please, look at Guerra et al. 2020 (https://www.nature.com/articles/s41467-020-17688-2) and Nunez et al. 2019 (https://besjournals.onlinelibrary.wiley.com/doi/full/10.1111/1365-2664.13319). 

The filling of gaps in knowledge, here the blind spots in the global scale studies, is an important part of build an appropriate global-scale databases.

I suggest you to check, which plant species of 44 tested in this study were were not previously tested for SLA, LL and LDMC. If your study provides a first data for a part of species/families/orders, it it a value of the manuscript. 

Round 2

Reviewer 1 Report

The manuscript ‘Leaf traits of trees in tropical dry evergreen forests of peninsular India’ by Udayakumar & Sekar maintains in this second version the positive points reported to the first version (good work of measuring and compiling relevant plant functional traits, promising approach to elucidate key processes, etc.). Moreover, it has been improved in several ways, chiefly in presenting some graphic synthetic results, introducing the CSR evaluation and perspective, and including extensive data on other ecosystems (new Table 2). Moreover, my concerns have been responded by the authors point by point, mostly (but not all) with appropriate changes. This is really a substantial improvement.

However, there are some drawbacks remaining (or even worsening) in this second version, which merit some improving actions from the authors:

  1. The discussion remains too long to my view. Although some added parts are due, there remains some redundancy with results, and within the same discussion, which should be avoided. The new Table 2 is a valuable addition; but some unusual values (to my view) included may indicate heterogeneous origin of the data (high values of SLA may correspond to herbaceous understory?). In general, the ecological context of the data reported should be acknowledged
  2. The conclusions, although deriving from the own results, are much more a summary of the discussion, not properly conclusions. They have been even enlarged from the first version.
  3. The list of references has even been enlarged (from 124 to 151), which seems too long.
  4. In the manuscript remain a number of bad sentences and typing mistakes, which must be clearly improved; this improving is particularly needed in the parts of the text added in this second version (including table heading and figure captions). Some if this improving include technical words bad-tipped (i.e., redaral instead of ruderal) and correcting data presentation (i.e., 32 °C, not 32° C; 0.29mm2, not 0.29mm2; and so on).
